# Trends in SARS-CoV-2 infection prevalence during England's roadmap out of lockdown, January to July 2021

**Oliver Eales**[1,2]*, **Haowei Wang**[1,2], **David Haw**[1,2], **Kylie E. C. Ainslie**[1,2,3], **Caroline E. Walters**[1,2], **Christina Atchison**[1], **Graham Cooke**[4,5,6], **Wendy Barclay**[4], **Helen Ward**[1,5,6], **Ara Darzi**[5,6,7], **Deborah Ashby**[1], **Christl A. Donnelly**[1,2,8], **Paul Elliott**[1,5,6,9,10,11]*, **Steven Riley**[1,2]*

**1** School of Public Health, Imperial College London, London, United Kingdom, **2** MRC Centre for Global infectious Disease Analysis and Abdul Latif Jameel Institute for Disease and Emergency Analytics, Imperial College London, London, United Kingdom, **3** Centre for Infectious Disease Control, National Institute for Public Health and the Environment, Bilthoven, The Netherlands, **4** Department of Infectious Disease, Imperial College London, London, United Kingdom, **5** Imperial College Healthcare NHS Trust, London, United Kingdom, **6** National Institute for Health Research Imperial Biomedical Research Centre, London, **7** Institute of Global Health Innovation at Imperial College London, London, United Kingdom, **8** Department of Statistics, University of Oxford, Oxford, United Kingdom, **9** MRC Centre for Environment and Health, School of Public Health, Imperial College London, London, United Kingdom, **10** Health Data Research (HDR) UK London at Imperial College, London, United Kingdom, **11** UK Dementia Research Institute at Imperial College, London, United Kingdom

* o.eales18@imperial.ac.uk (OE); p.elliott@imperial.ac.uk (PE); s.riley@imperial.ac.uk (SR)

## Abstract

### Background

Following rapidly rising COVID-19 case numbers, England entered a national lockdown on 6 January 2021, with staged relaxations of restrictions from 8 March 2021 onwards.

### Aim

We characterise how the lockdown and subsequent easing of restrictions affected trends in SARS-CoV-2 infection prevalence.

### Methods

On average, risk of infection is proportional to infection prevalence. The REal-time Assessment of Community Transmission-1 (REACT-1) study is a repeat cross-sectional study of over 98,000 people every round (rounds approximately monthly) that estimates infection prevalence in England. We used Bayesian P-splines to estimate prevalence and the time-varying reproduction number ($R_t$) nationally, regionally and by age group from round 8 (beginning 6 January 2021) to round 13 (ending 12 July 2021) of REACT-1. As a comparator, a separate segmented-exponential model was used to quantify the impact on $R_t$ of each relaxation of restrictions.

**Data Availability Statement:** Access to individual level REACT-1 data is restricted due to ethical and security considerations. Researchers wishing to inquire about access to individual level data should

email react.access@imperial.ac.uk. Summary statistics, code and data, including the weighted daily number of positive tests and daily total number of tests, are available at the github repository: mrc-ide/reactidd (https://doi.org/10.5281/zenodo.6557251). Additional summary statistics and results from the REACT-1 programme are also available at https://www.imperial.ac.uk/medicine/research-and-impact/groups/react-study/real-time-assessment-of-community-transmission-findings/. REACT-1 study materials are available for each round at https://www.imperial.ac.uk/medicine/research-and-impact/groups/react-study/for-researchers/react-1-study-materials/. Computer code supporting the paper is available at https://github.com/mrc-ide/reactidd and also on zenodo (DOI:10.5281/zenodo.7085123).

**Funding:** The REACT-1 study was funded by the Department of Health and Social Care in England. SR, CAD acknowledge support: Medical Research Council (MRC) Centre for Global Infectious Disease Analysis, National Institute for Health Research (NIHR) Health Protection Research Unit (HPRU), Wellcome Trust (200861/Z/16/Z, 200187/Z/15/Z), and Centres for Disease Control and Prevention (US, U01CK0005-01-02). GC is supported by an NIHR Professorship. HW acknowledges support from an NIHR Senior Investigator Award and the Wellcome Trust (205456/Z/16/Z). PE is Director of the MRC Centre for Environment and Health (MR/L01341X/1, MR/S019669/1). PE acknowledges support from Health Data Research UK (HDR UK); the NIHR Imperial Biomedical Research Centre; NIHR HPRUs in Chemical and Radiation Threats and Hazards, and Environmental Exposures and Health; the British Heart Foundation Centre for Research Excellence at Imperial College London (RE/18/4/34215); and the UK Dementia Research Institute at Imperial (MC_PC_17114). We thank The Huo Family Foundation for their support of our work on COVID-19. The funders had no role in study design, data collection and analysis, decision to publish, or preparation of the manuscript.

**Competing interests:** The authors have declared that no competing interests exist.

## Results

Following an initial plateau of 1.54% until mid-January, infection prevalence decreased until 13 May when it reached a minimum of 0.09%, before increasing until the end of the study to 0.76%. Following the first easing of restrictions, which included schools reopening, the reproduction number $R_t$ increased by 82% (55%, 108%), but then decreased by 61% (82%, 53%) at the second easing of restrictions, which was timed to match the Easter school holidays. Following further relaxations of restrictions, the observed $R_t$ increased steadily, though the increase due to these restrictions being relaxed was offset by the effects of vaccination and also affected by the rapid rise of Delta. There was a high degree of synchrony in the temporal patterns of prevalence between regions and age groups.

## Conclusion

High-resolution prevalence data fitted to P-splines allowed us to show that the lockdown was effective at reducing risk of infection with school holidays/closures playing a significant part.

## Author summary

Throughout the COVID-19 pandemic in England there has been a high rate of testing. However, the case data obtained from this mass testing is prone to many biases due to changing testing rates and behaviours. The REal-time Assessment of Community Transmission-1 (REACT-1) study instead tests random subsets of the population of England for SARS-CoV-2, providing a relatively unbiased signal of infection prevalence over time. Here we present the temporal analysis of rounds 8–13 of REACT-1, running from January to July 2021. During this period a national lockdown was introduced in England, followed by the staged relaxation of restrictions. We find that the lockdown was highly effective at reducing levels of infection prevalence in England, with prevalence declining until mid-May. However, as restrictions were gradually relaxed the reproduction number, R, increased to greater than 1 (the threshold for epidemic growth) and infection prevalence once more entered a phase of growth. Analysis of the step-changes in R after each restriction relaxation highlighted the significant effect that school holidays/closures likely had on R over this period. Additionally, we found that increases in R were likely offset by the high rates of vaccination that were achieved by July 2021.

## Introduction

Throughout the SARS-CoV-2 pandemic, non-pharmaceutical interventions (NPIs) have been crucial in controlling the spread of the virus, and have been highly effective [1–3]. NPIs aim at reducing the number of social contacts an individual makes, therefore severing possible links of transmission. A wide variety of NPIs have been introduced globally including stringent restrictions such as lockdowns [1] and school closures [4]. Since the development of effective vaccines [5,6], vaccination has been at the forefront of public health interventions in many populations. However, NPIs will remain highly important and valuable tools in the future and it is crucial to understand their effect on transmission in planning for possible future waves of severe SARS-CoV-2 variants and future emerging pathogens.

On 6 January 2021, following record high numbers of cases, England entered a national lockdown [7]. Over the following months, the restrictions were gradually eased with phased reopenings [8]. The first restriction easing (step 1a) on 8 March saw the return of face-to-face teaching in schools (previously schools were closed for most students in England) and colleges and allowed limited contact with individuals from different households outdoors (up to 2 people). The second easing (step 1b) occurred on 29 March and expanded the allowed outdoor social contacts with individuals from different households (up to 6 people). On 14 April the third restriction easing (step 2) occurred allowing indoor non-essential retail and outdoor hospitality to reopen. The dates of step 1b and step 2 aligned closely with the Easter holidays that saw many schools closed for the first two weeks of April. The fourth easing (step 3) on 17 May allowed indoor hospitality to reopen for groups of up to 6 people. The final easing of restrictions was scheduled to occur on 21 June and removed nearly all domestic restrictions, but due to rising cases was postponed to 19 July [9]. More detailed descriptions of the changes at each step are given in Table 1 [8].

The REal-time Assessment of Community Transmission—1 (REACT-1) study is a repeat cross-sectional study that estimated the prevalence of SARS-CoV-2 infections in England approximately every month from May 2020 to March 2022 [10]. Due to the random sampling procedure, the study reduces many of the biases present in symptomatic universal testing programs [11]. Here we present the inferred trends in infection prevalence and the time-varying reproduction number $R_t$ during the period of the roadmap out of the third lockdown in England, from 6 January to 12 July 2021 both nationally and for sub-groups (regions and approximate age-quartiles). We use prevalence to describe the effect that each step down in restrictions had on the reproduction number.

## Methods

### Ethics

The REACT-1 study received research ethics approval from the South Central-Berkshire B Research Ethics Committee (IRAS ID: 283787).

### REACT-1 study protocol

The REACT-1 study protocol has been described in detail elsewhere [12]. During each round of the study a random subset of the population (over 5 years old) of England, was chosen from

**Table 1. Summary table of changes in restrictions over the period of rounds 8–13 of the REACT-1 study [8].**

| Restriction step | Date | Main changes |
|---|---|---|
| Lockdown | 06/01/2021 | • Stat at home rule announced limiting the reasons people can leave their homes<br>• All primary schools, secondary schools and colleges close, moving to remote learning<br>• All non-essential retail, hospitality, sports facilities and entertainment venues close<br>• People advised to work from home unless they cannot reasonably do so |
| Step 1a | 08/03/2021 | • Face-to-face education in schools and colleges to resume<br>• Practical courses at English universities can resume<br>• People allowed to leave home for recreation and exercise outdoors one-to-one with people not part of their household |
| Step 1b | 29/03/2021 | • Outdoor gatherings of either 6 people (rule of 6) or 2 households to be allowed<br>• Outdoor sports facilities to be allowed to reopen<br>• 'Stay at home' rule will end but people should continue to work from home where they can |
| Step 2 | 12/04/2021 | • Non-essential retail allowed to reopen<br>• Outdoor attractions and settings including outdoor hospitality venues, zoos and theme parks allowed to reopen<br>• Wider social contact rules (e.g rule of 6) will still apply in all settings |
| Step 3 | 17/05/2021 | • Indoor hospitality will be allowed to reopen. However customers will have to order eat and drink while seated.<br>• Other indoor locations such as cinemas and children's play areas will also be allowed to reopen<br>• Rule of 6 or 2 households to apply indoors, but not outdoors where gatherings will be allowed for up to 30 people |

the national health service (NHS) general practitioner's list and invited to participate in the study. Individuals who agreed to participate provided a self-administered throat and nose swab (parent/guardian administered for those aged 5–12 years old) that underwent reverse transcription polymerase chain reaction (rt-PCR) testing to determine the presence of SARS-CoV-2. There was a round of the study approximately monthly from May 2020 to March 2022, with rounds 8 to 13 of the study running from 6 January to 12 July 2021 with between 98,233 and 167,642 swab tests performed each round (S1 Table). There was a steady decline in the percentage of individuals responding to the initial invitation, decreasing from 22.0% in round 8 to 11.7% in round 13. During the first 11 rounds the study protocol aimed to achieve approximately equal sample sizes from each lower tier local authority (LTLA, N = 315), whereas in rounds 12 and 13 the study protocol changed so that sample sizes were representative of each LTLA's population size. Participants were assigned individual weights using rim weighting [13] by: sex, deciles of the IMD, LTLA counts and ethnic group. This ensured the results of any analysis were more representative of the population of England as a whole even between rounds in which the study protocol changed. Positivity in the study was defined as rt-PCR tests with an N-gene Ct value less than 37 or with both N- and E-gene detected.

## Bayesian P-spline model

A Bayesian Penalised spline (P-spline) model was fitted to the daily weighted prevalence in order to obtain continuous estimates of the expected infection prevalence. The Bayesian P-spline model has been described in detail elsewhere [14]. In short, the entire time-series is split into equal sized knots of approximately 5 days (high enough density to prevent underfitting), with 3 further knots defined beyond both the beginning and end of the time series in order to reduce edge effects. Fourth-order basis splines (b-splines) are defined across all knots. The P-spline model consists of a linear combination of these b-splines,

$$g(P_t) = \sum_{i=1}^{N} b_i B_{i,t},$$

where $g()$ is the link function (logit function for binomial data), $P_t$ is the prevalence on day t, $B_{i,t}$ is the value of the $i^{th}$ b-spline on day t, $b_i$ is the $i^{th}$ b-spline coefficient, and the summation is over all N b-splines. Overfitting is prevented through the inclusion of a second-order random-walk prior distribution on the b-spline coefficients, $b_i = 2b_{i-1}-b_{i-2}+u_i$, where the random error $u_i$ is normally distributed, $u_i \sim N(0, \rho)$. The first and second b-spline coefficients are given an uninformative constant prior distribution, $b_1, b_2 \sim Constant$. The prior distribution's parameter $\rho$ penalises changes in the first derivative of the link function, effectively penalising changes in the growth rate. The parameter $\rho$ was given an uninformative inverse gamma prior distribution, $\rho \sim IG(0.001, 0.001)$.

We fit the model to the national daily weighted number of positive tests and weighted total number of tests assuming a binomial likelihood. Fitting was performed using a No-U Turns Sampler (NUTS) [15] implemented in STAN [16]. An analogous model was fit to data for each region of England (North West, North East, Yorkshire and The Humber, West Midlands, East Midlands, East of England, London, South West, South East) and for four approximate age-quartiles (5–17, 18–34, 35–54, 55+ years). However, in this analogous model we assume a constant value for $\rho$ estimated from the model fit to the national data. Continuous estimates of the time-varying reproduction number, $R_t$, were estimated from the national and regional prevalence estimates [14] under the assumptions that $R_t$ was constant over the previous two week period and that the generation time (the average time between infections of a primary case and one of its secondary cases) followed a gamma distribution with shape parameter = 2.29, and rate parameter = 0.36 [17], equivalent to a mean of 6.36 days and a standard deviation of 4.20

days. As a sensitivity analysis we also estimated $R_t$ using a generation time following a gamma distribution with shape parameter = 2.20, and rate parameter = 0.48, equivalent to a mean of 4.58 days and a standard deviation of 3.09 days. These parameters were estimated for the Delta variant's generation time [18] and so show how our $R_t$ estimates during the period when the Delta variant emerged might have been biased. Continuous estimates of the instantaneous exponential growth rate were estimated from the estimates of prevalence by age group [14]. We chose not to estimate independent reproduction numbers for individual age groups because of their interconnected transmission dynamics [19].

## Segmented-exponential model

In order to quantify the effect of the various levels of restrictions on the reproduction number, we fit a Bayesian segmented-exponential model. The period of the study was split into six distinct periods with breakpoints at 6 January, 8 March, 29 March, 14 April and 16 May 2021 – the dates of key restriction changes–with an additional time-delay parameter, $\tau$. The prevalence on day t+1 was then determined by the equation:

$$P(t + 1) = P(t) \times e^{r(t)}$$

where the growth rate on day t, $r(t)$, given by:

$$r(t) = r_0, \qquad t < 6\ Jan + \tau$$

$$= r_1, \qquad 6\ Jan + \tau \leq t < 8\ Mar + \tau$$

$$= r_2, \qquad 8\ Mar + \tau \leq t < 29\ Mar + \tau$$

$$= r_3, \qquad 29\ Mar + \tau \leq t < 14\ Apr + \tau$$

$$= r_4, \qquad 14\ Apr + \tau \leq t < 16\ May + \tau$$

$$= r_5, \qquad t \geq 16\ May + \tau$$

The parameters $r_0$, $r_1$, $r_2$, $r_3$, $r_4$ and $r_5$ are the growth rates during each period of time between changes in restrictions with a time offset parameter, $\tau$, introducing a delay between changes in restrictions and a corresponding change in the growth rate. The resulting 6 growth rate parameters were assumed to have an uninformative constant prior distribution. The time-delay parameter was given a uniform prior distribution from 0 to 14 days, $\tau \sim U(0,14)$. The initial prevalence on the first day for which we have data (30 December 2020) was also given an uninformative constant prior distribution from 0 to 1.

We fit the model to the daily weighted number of positive tests and weighted total number of tests assuming a binomial likelihood for national data and for subgroups (9 regions and approximate age quartiles). Model fitting was again performed using NUTS [15] as implemented in STAN [16].

A further model was also fitted to just the national data. The model is the same as the model described above, but also allowed for changes to the daily growth rate based on the proportion of individuals in England vaccinated with 1 dose (more than 21 days before), the proportion vaccinated with 2 doses (more than 14 days before) [20] and the proportion of cases associated with the Delta variant [21]. This refined model could not be fit to subgroups due to smaller sample sizes and data availability for vaccination rates/ Delta variant proportions. The

prevalence on day t+1 in this model is given by:

$$P(t+1) = P(t) \times e^{r(t)} \times (1 - V_1(t) - V_2(t) + \gamma_1 V_1(t) + \gamma_2 V_2(t)) \times (1 - D(t) + \epsilon D(t))$$

As before, $r(t)$ takes a constant value over each period of time between restrictions. However, there are now two extra bracketed terms that modify its effect, with the first adjusting for: the proportion of the population who had received their second vaccine dose (more than 14 days before) on day t, $V_2(t)$; and the proportion who had only received one vaccine dose (more than 21 days before), $V_1(t)$. Hence, the parameters $\gamma_1$ and $\gamma_2$, which are given uniform prior distributions $\gamma_1 \sim U(0,1)$ and $\gamma_2 \sim U(0,\gamma_1)$, reduce the apparent daily growth of prevalence dependent on the proportion vaccinated.

The second bracketed term in the equation allows adjustment based on the proportion of positive samples that were determined to be the Delta variant, $D(t)$. The parameter, which is given a uniform prior distribution, $\epsilon \sim U(1,5)$, increases the apparent daily growth of prevalence as Delta increases in proportion. The proportion of Delta was reported weekly, and daily values were calculated assuming a linear relationship between each pair of weeks. The parameters $\gamma_1$, $\gamma_2$ and $\epsilon$ effectively adjust the intrinsic daily growth rate r for vaccination or the emergence of Delta. The apparent daily growth rate, $r_A(t)$, that is the growth rate that is actually observed can be calculated through the relationship:

$$e^{r_A(t)} = e^{r(t)} \times (1 - V_1(t) - V_2(t) + \gamma_1 V_1(t) + \gamma_2 V_2(t)) \times (1 - D(t) + \epsilon D(t))$$

Estimates of all growth rates are converted into estimates of the reproduction number, R, again assuming a gamma-distributed generation time with shape parameter = 2.29, and rate parameter = 0.36 [17] through the equation $R = \left(1 + \frac{r}{b}\right)^n$ [22].

## Results

### Trends in infection prevalence

Infection prevalence exhibited a u-shape during the period of this study (Fig 1). Within rounds 8 to 13 inclusive, we estimated prevalence to be highest nationally in January 2021 with a value of 1.54% (1.44%, 1.64%) on 6 January (first official day of round 8). Infection prevalence plateaued until mid-January, after which prevalence fell sharply, reaching a minimum (Fig 2C) of 0.09% (0.06%, 0.11%) on 13 May (20 April, 21 May). After this date, national infection prevalence increased steadily, reaching a value of 0.74% (0.59%, 0.91%) on 12 July (last day of round 13).

Although initial prevalence varied between regions, subsequent trends were more consistent. There was a high level of variation in the modelled infection prevalence in January with infection prevalence on 6 January highest in London at 2.85% (2.51%, 3.22%) and lowest in Yorkshire and The Humber at 0.80% (0.63%, 0.99%) (Figs 2A and S1). However, regional trends in infection prevalence showed a large degree of synchrony, with infection prevalence in most regions decreasing and reaching a minimum (Fig 2C) at approximately the same time as the national estimate, with only an indication that the South East reached its minimum earlier (with wide credible intervals). There was less variation regionally at the end of the study period (12 July) with infection prevalence highest in Yorkshire and The Humber at 1.25% (0.76%, 1.97%) and lowest in the South East at 0.44% (0.25%, 0.73%) at that time.

Modelled infection prevalence varied by age group (Figs 2B and S2) with the youngest age group, 5–17 year olds, having the highest infection prevalence at the beginning and end of the study period at 2.38% (2.04%, 2.77%) and 1.33% (0.93%, 1.86%) respectively. In contrast, the oldest age group, 55+ year olds, had the lowest infection prevalence at 1.06% (0.94%, 1.18%)

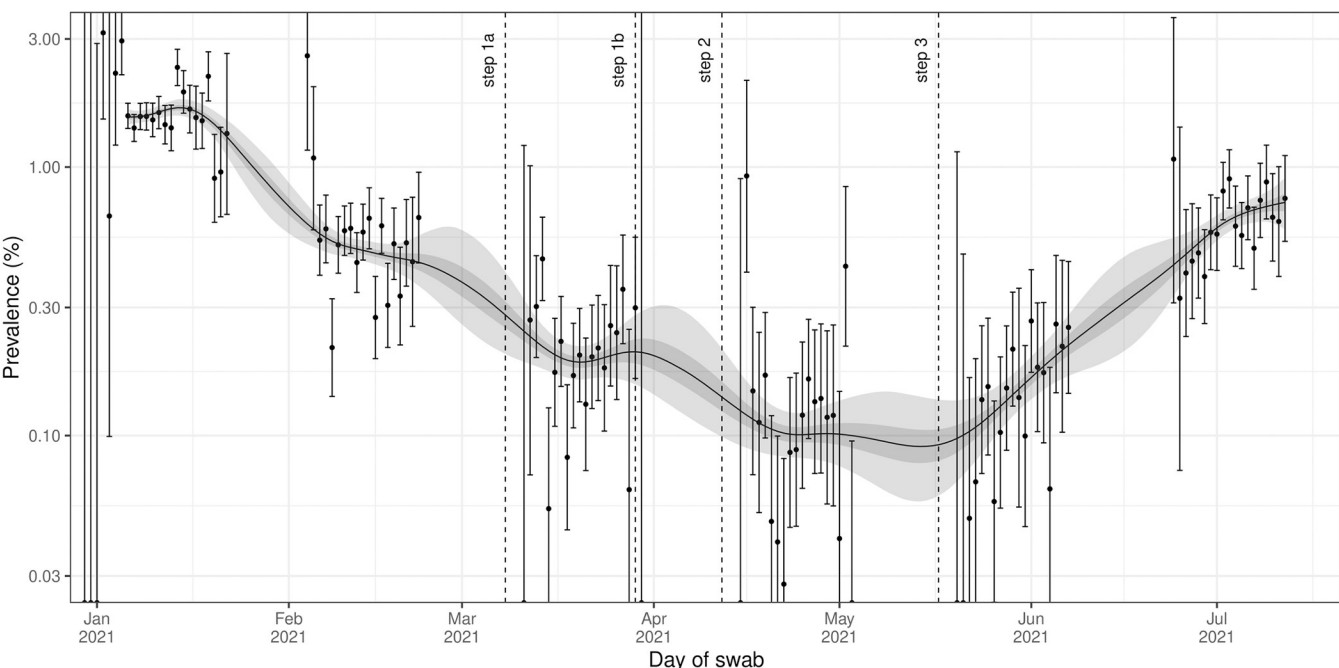

**Fig 1. Smoothed estimates of infection prevalence in England.** Modelled infection prevalence in England from 6 January to 12 July 2021 estimated using a Bayesian P-spline model fit to rounds 1–13 of REACT-1 data (only shown for rounds 8–13). Dashed lines show the date of key restriction changes in England. Estimates of infection prevalence are shown with a central estimate (solid line) and 50% (dark shaded region) and 95% (dark shaded region) credible intervals. Daily weighted estimates of swab positivity (points) are shown with 95% confidence intervals (error bars). Dashed lines show the date of key restriction changes in England.

on 6 January and 0.34% (0.20%, 0.54%) on 12 July. Trends in infection prevalence again showed a large degree of synchrony between age groups with infection prevalence reaching a minimum (Fig 2C) at approximately the same time. However, infection prevalence was observed to begin decreasing from 6 January in 5–17 year olds, whereas in the other age groups infection prevalence only began decreasing in mid-January.

## Trends in reproduction number and growth rate

National estimates of the time-varying reproduction number, $R_t$, showed clear temporal features (Fig 3). Temporary increases in $R_t$ were seen in February and in March, with a gradual increase in $R_t$ observed from mid-April onwards. $R_t$ plateaued at a high of approximately 1.25 for the entirety of June, before declining slightly into July. Under the assumption of a shorter generation time, more appropriate for the Delta variant (see Methods), $R_t$ showed the same temporal patterns though but was slightly lower in value (S3 Fig). This suggests our $R_t$ estimates may be slightly inflated during the period when Delta was the main variant. Regional trends in $R_t$ were broadly similar, but London saw clearer temporal features (S4 Fig). In June $R_t$ was greatest in London, reaching 1.56 (1.29, 1.85) on 16 June before rapidly decreasing to 0.88 (0.63, 1.21) on 12 July, the lowest regional $R_t$ on this date. Similar trends in $R_t$ were again observed when a shorter generation time, more appropriate for the Delta variant, was used (S5 Fig). Estimates in the instantaneous growth rate between age groups showed similar overall patterns (Fig 4). However, during March there was a clear increase in growth rate followed by a decrease into April for 5-17- and 18–34-year-olds; this was not observed in 35–54 and 55 + year-olds which showed little variation in growth rate over this period.

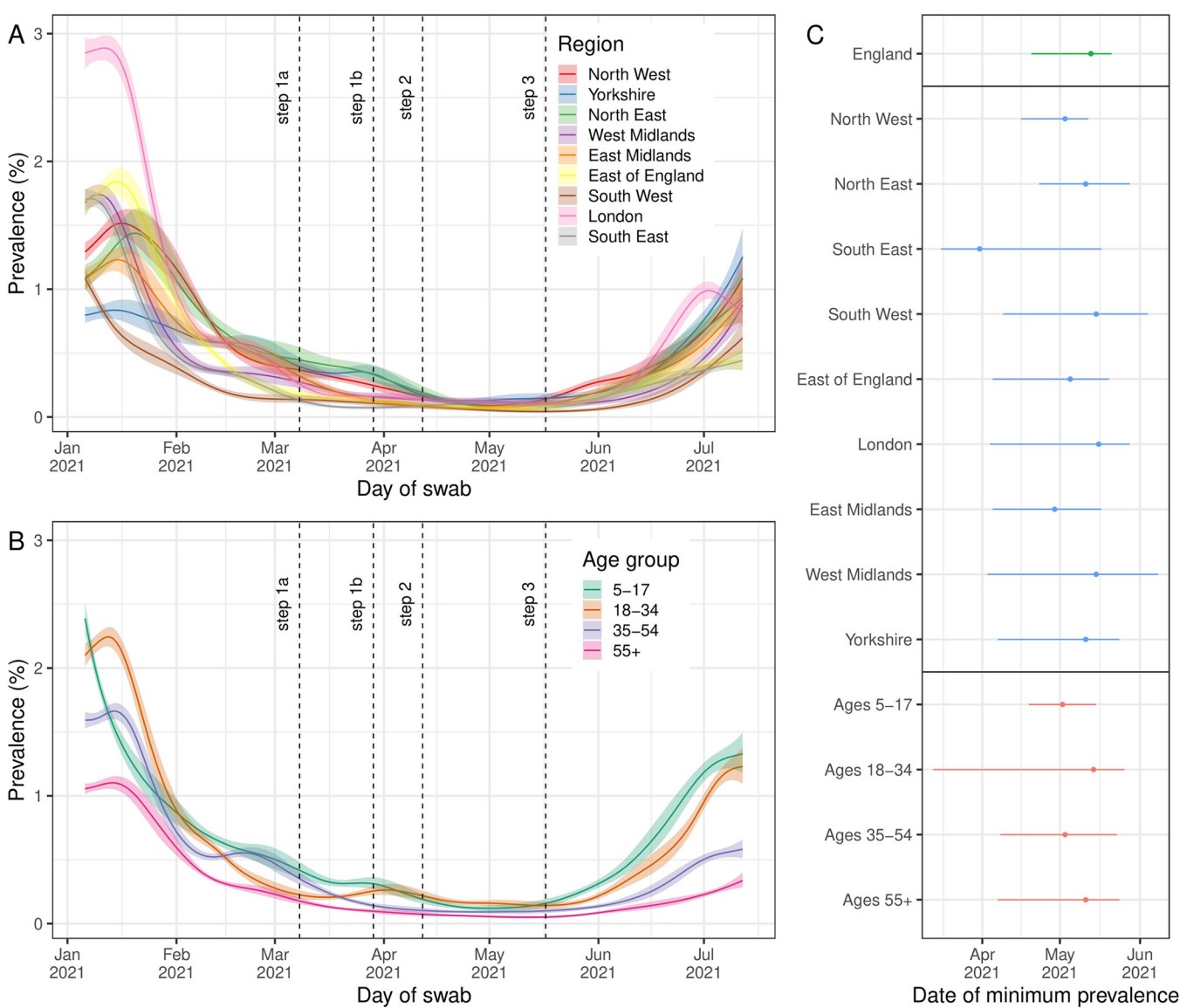

**Fig 2. Differences in smoothed estimates of infection prevalence by region and age group. (A)** Regional infection prevalence from 6 January to 12 July 2021 estimated using a Bayesian P-spline model fit to all 13 rounds of REACT-1 (only shown for rounds 8–13) assuming a constant second-order random-walk prior distribution (value set to estimate from national model fit). In the legend Yorkshire is short for Yorkshire and The Humber. **(B)** Infection prevalence estimates by age group from 6 January to 12 July 2021 estimated using a Bayesian P-spline model fit to all 13 rounds of REACT-1 (only shown for rounds 8–13) assuming a constant second-order random-walk prior distribution (value set to estimate from national model fit). All estimates of infection prevalence are shown with a central estimate (solid line) and 50% credible interval (shaded region). Full data and 95% credible intervals are shown in supplementary Figs 1 and 2. Dashed lines show the date of key restriction changes in England. **(C)** The inferred date of minimum prevalence from 6 January to 12 July 2021 for all models fit to national prevalence (green), regional prevalence (blue) and prevalence by age group (red) with median (point) and 95% credible intervals (line) shown.

## Quantifying the effect of non-pharmaceutical interventions

In order to quantify the effects of the NPIs we fitted a segmented-exponential model to daily infection prevalence. We found a time delay between a change in NPIs and a corresponding change in the reproduction number, R, of 8 (7, 10) days (S2 Table). Before the introduction of the lockdown, R was estimated at 1.08 (1.00, 1.19), falling to 0.76 (0.75, 0.78) after it was introduced, corresponding to a multiplicative change of 0.71 (0.63, 0.78) (Fig 5A, S3 Table). After

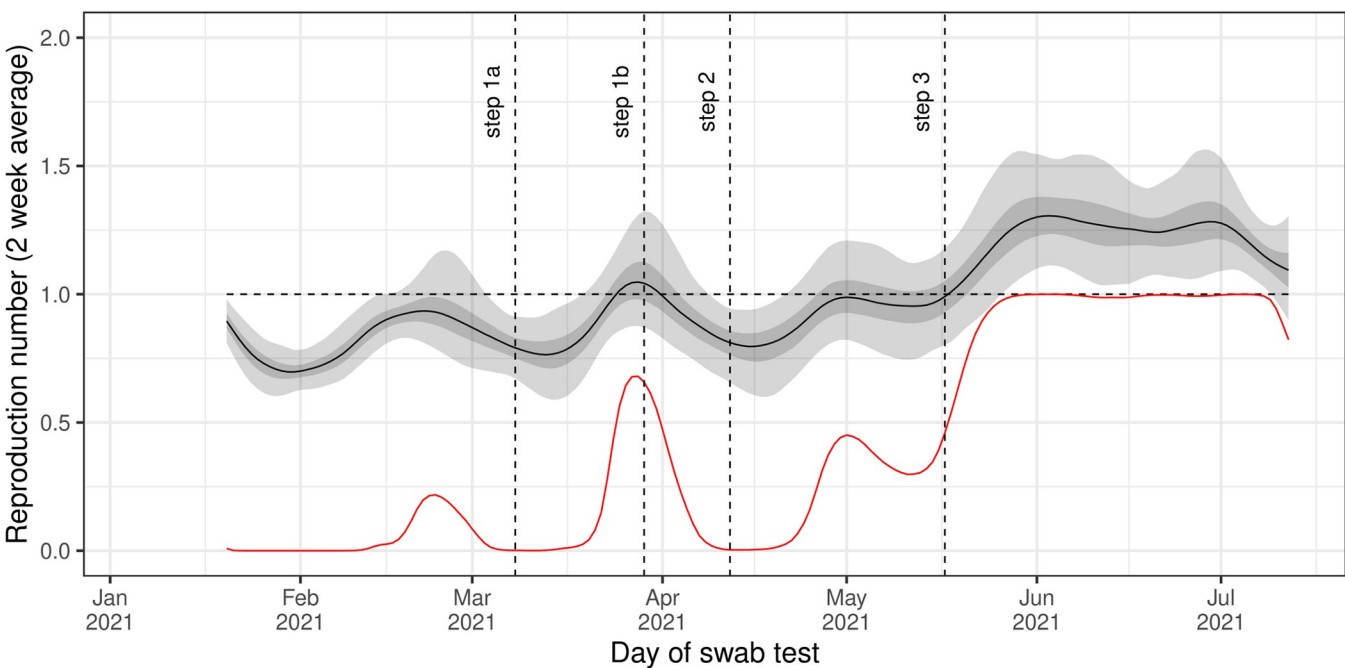

**Fig 3. Trends in the time-varying reproduction number.** Rolling two-week average (averaged over prior two weeks) reproduction number as inferred from the Bayesian P-spline model fit to all data assuming a gamma distributed generation time with shape parameter = 2.29, and rate parameter = 0.36. Estimates of the reproduction number are shown with a central estimate (solid line) and 50% (dark shaded region) and 95% (light shaded region) credible intervals. The red line shows the probability that R>1 over time. Vertical dashed lines show the dates of key changes in restrictions. Horizontal dashed line shows R = 1 the threshold for epidemic growth.

step 1a R increased to 1.39 (1.20, 1.57), an increase by a factor of 1.82 (1.55, 2.08). However, after step 1b R dropped below one to 0.39 (0.28, 0.57), a multiplicative reduction by a factor of 0.28 (0.18, 0.47). After step 2 and step 3 R increased to 1.07 (1.03, 1.11) and 1.28 (1.24, 1.30) respectively reflecting multiplicative growth by factors of 2.71 (1.84, 3.85) followed by 1.20 (1.13, 1.26).

Regional patterns in multiplicative changes (S6 Fig and S3 Table) were consistent with the national trends with the exception of the South West which showed an increase in R after the lockdown was introduced, though this likely reflects inaccuracy in the R estimate for the period prior to the lockdown for which we have little data.

Patterns for each age group were also consistent with the national trends (S6 Fig, and S3 Table), with the exception of there being no significant reduction in R brought about by the introduction of lockdown for 5–17 year olds. There was some variation between age groups with step 1a leading to a greater multiplicative increase in R for 18–34 year olds at 2.40 (1.78, 3.19) compared to 35–54 year olds at 1.06 (0.66, 1.57). The time-delay parameter showed far greater variation for subgroup analysis with some sub groups having credible intervals including the upper bound of the prior distribution.

## Correcting for vaccine effectiveness and delta

Including additional effects in the segmented-exponential model that account for the proportion vaccinated (one or two doses) and the proportion of infections caused by Delta (Fig 5B), produced similar estimates of R for each period (though it was no longer constant for each period). Estimates of the intrinsic R for the Alpha variant (the estimated R had vaccination and Delta not influenced transmission) were similar to the actual estimates of R for the periods

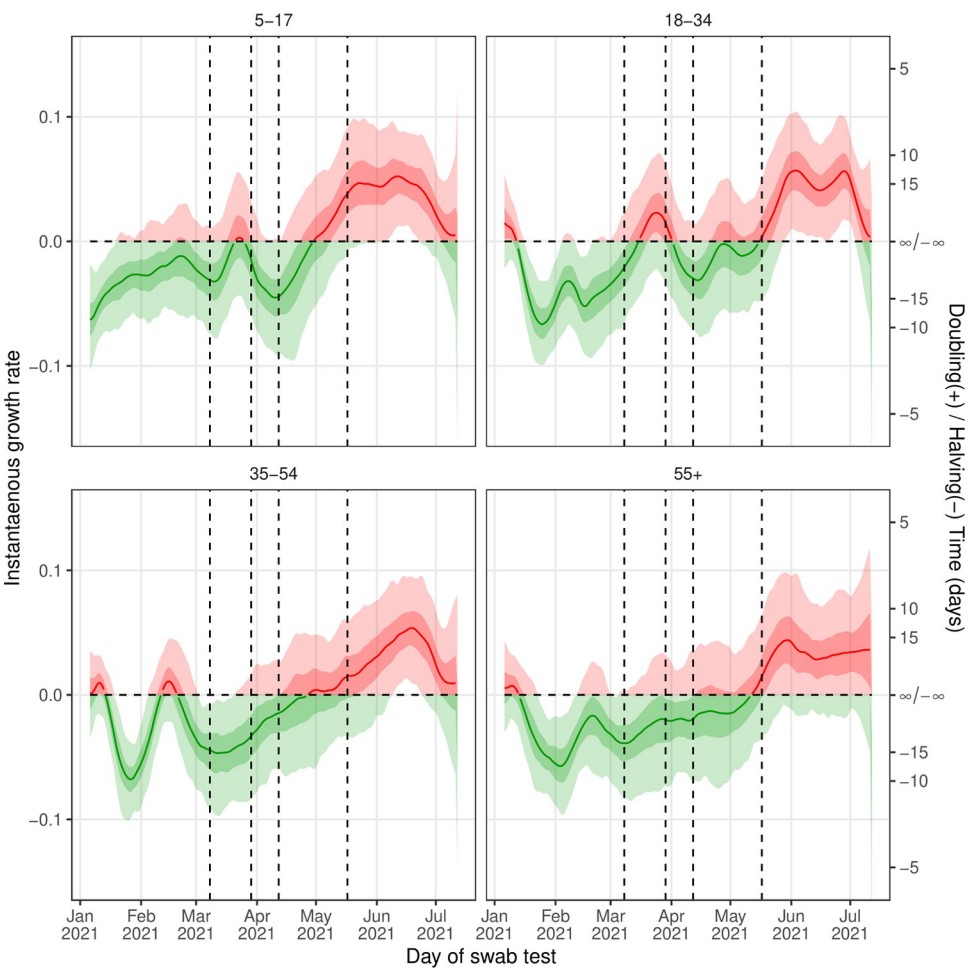

**Fig 4. Trends in the instantaneous growth rate by age group.** Instantaneous growth rate as inferred from the Bayesian P-spline models fit to data for each age group. Estimates of the instantaneous growth rate are shown with a central estimate (solid line) and 50% (dark shaded region) and 95% (light shaded region) credible intervals. Estimates are colored by whether their value is greater than 0 (red) or less than 0 (green). Vertical dashed lines show the dates of key changes in restrictions. The horizontal dashed line shows growth rate = 0 the threshold for epidemic growth. The right hand axis gives the corresponding doubling / halving times for a given growth rate.

of time before step 2 (S3 Table), reflecting very little effect due to single-dose vaccinations being detected. The multiplicative increase in intrinsic R due to step 2 and step 3 were 2.85 (1.61, 4.04) and 1.49 (1.02, 1.75) respectively. The estimates of the intrinsic R after step 2 were larger than estimates for the actual observed R, but with wider credible intervals.

## Discussion

We have presented the trends in infection prevalence and $R_t$ in England from January to July 2021. Accurate measurements of infection prevalence are of interest to both the individual, reflecting the risk of a social contact being infected with SARS-CoV-2, and the population as a whole, determining what restrictive measures are needed to reduce the burden of SARS-CoV-2 within the population. The third national lockdown introduced on 6 January was found to be effective at keeping $R_t$ below one, but by June, after numerous restrictions had been eased $R_t$ was greater than one and the pandemic was again in a phase of growth. Transient increases in $R_t$ were observed approximately at the time of step 1a, step 2 and step 3 of the restriction

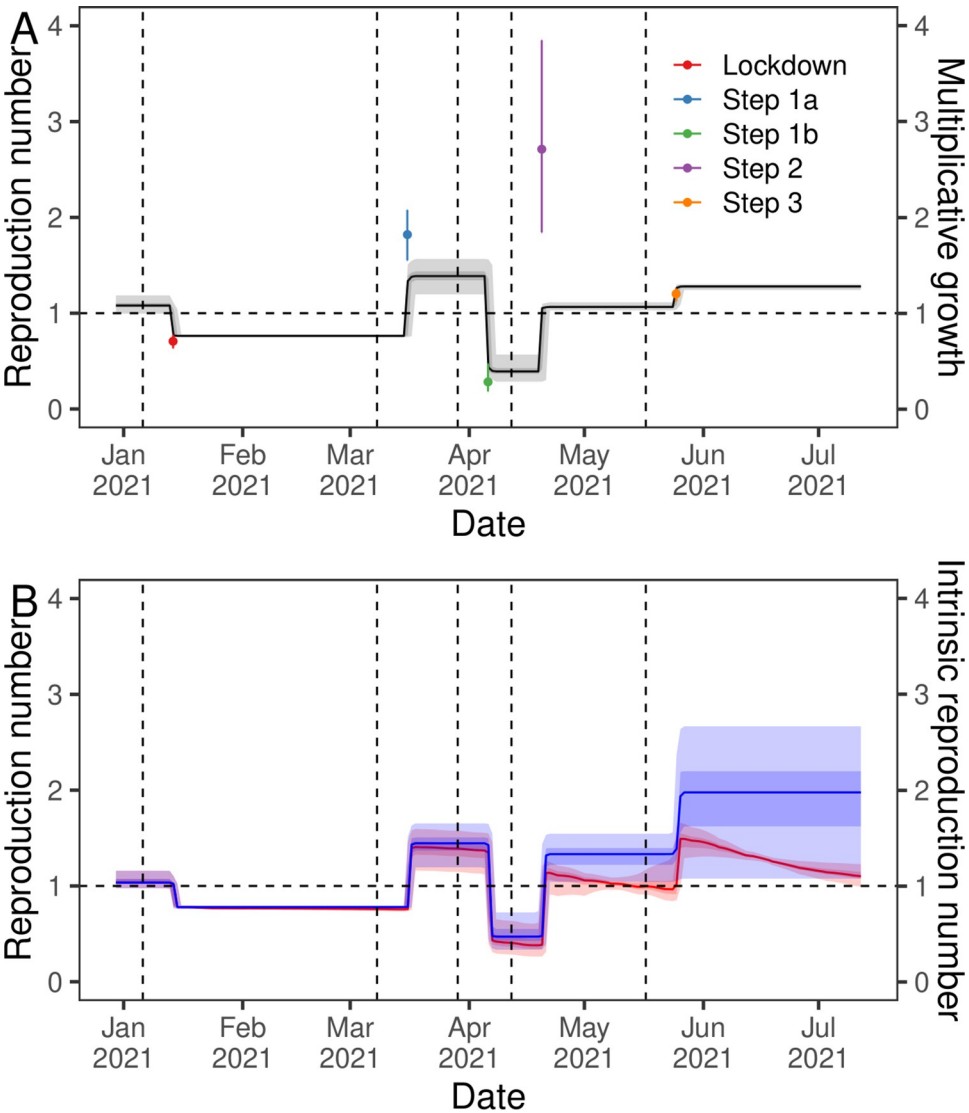

**Fig 5. Estimated discrete changes in the reproduction number. (A)** Inferred R over time (black line, left-axis) from the segmented-exponential model fit to rounds 8 to 13 of REACT-1. Also shown is the multiplicative growth in R due to each step change in restrictions (points, right-axis) and their 95% credible intervals (error bars). **(B)** Inferred R over time (red, left-axis) from the segmented-exponential model including vaccine and Delta fixed effects fit to rounds 8 to 13 of REACT-1. Also shown is the intrinsic R (R if vaccine and Delta effects excluded) over time (blue, right-axis). All estimates of R are shown with a central estimate (solid line) and 50% (dark shaded region) and 95% (light shaded region) credible intervals. Vertical dashed lines show the dates of key changes in restrictions. Horizontal dashed line shows R = 1 the threshold for epidemic growth.

easing and during February. This temporary increase in $R_t$ in England during February has not been previously reported. A decrease in $R_t$ was seen at the time of step 1b through to step 2. However, this coincided with the Easter holidays, a national holiday that resulted in many schools being closed for two weeks (some schools closed for three weeks) with many work-places closed over a long weekend; changes in behaviour during this period could be identified by temporary reductions in the mean number of social contacts [23]. There was a large degree of synchrony in the trends between regions over the entire period with no observed differences even during the emergence of the far more transmissible Delta variant which was introduced

earlier in the North West and London [24,25]. The regional synchrony in prevalence trends during Delta's emergence contrasts higher levels of asynchronous behaviour seen during the emergence of the Omicron variant in England [10]. This suggests that the changing levels of restrictions more heavily influenced the observed trends in infection prevalence over this period. Differences in the trends of age-group specific growth rates may highlight the effect the different restrictions had; step 1a, saw a large increase in growth rate for 5–17 and 18–34 year olds, two age-groups which contain school-aged/college-aged individuals.

Potential changes in the reproduction number at the time of key changes in restrictions were quantified using a segmented-exponential model. Step 1a in which schools were reopened led to a large increase in R. The effect of school closures on SARS-CoV-2 transmission has been well documented [26,27] though some studies have suggested measures of the effect are largely inflated by other correlated changes [28]. We are unable to rule this out using our dataset as step 1a also relaxed some rules on socialising between households. It is also possible that behaviours changed due to lockdown fatigue, compounding any sudden increases in R. This could explain why we measure an effect larger than other previously reported estimates for school closures [26], though our estimates are consistent with predictions based on social contact surveys in England for the same period of time as the study [29].

A large decrease in R was observed between step 1b and step 2 (approximately the Easter holidays) to a value lower than that observed before schools opened, most likely highlighting the combined effect of school closures and some work closures. Step 2 and step 3 led to clear increases in R reflecting the greater levels of social contacts driving increased transmission rates. However, even accounting for potential confounding effects due to vaccination and the emergence of Delta, the suggested intrinsic R values obtained were still comparable to the R value between step 1a and step 1b, when we may have expected it to be greater. This potentially be explained by a reduction in R due to seasonality [30], or the depletion of susceptible individuals in the population. Another potential explanation is that the estimate pre-Easter was inflated due to other temporary confounding effects.

Though REACT-1 should provide a relatively unbiased sample, it is possible that there are unknown biases that we are unable to account for. We attempted to account for the changes in sampling procedure between round 11 and 12 by using weighted values for our analyses, but the change may have introduced a bias in our estimates between these two rounds. Steadily decreasing participation rates over the study may also reflect an unknown time-dependent bias. Though, despite these unknowns, it seems likely that the REACT-1 data continued to suffer from less bias than routine surveillance [11].

During England's roadmap out of lockdown, though restrictions changed on specific days, behaviour changed more gradually [23]. The Bayesian P-spline model allows for these gradual changes with smooth changes in the growth rate. The model is highly informative, providing smooth estimates of prevalence and $R_t$, but does not allow the effect of restriction changes to easily be quantified, and possible step changes in growth rate due to restriction changes would be smoothed out. The segmented exponential model allows the effect of each change in restriction to easily be quantified, assuming that the growth rate only undergoes step changes at the time of restriction changes. The estimated growth rates (and associated Rs) are averages over the periods between restrictions. This may bias the model if there are any significant changes in growth rate between restriction easings; a gradual increase in growth rate over a long period would be identified as step changes at the date of any restriction changes. Due to the limited nature of the data, the model assumed a constant time delay between a restriction changing and R changing. A constant delay is valid if it is just due to the delay between incidence and infection prevalence, but it is also possible that behavioural changes, which may vary between restrictions, also had an influence on the delay. We found a time delay of 8 days which is

approximately consistent with the delay observed between incidence and prevalence in previous studies [31]. Additional constant effects were included in the model to adjust for vaccination and Delta's emergence, whilst considering the effect of restriction changes. However, it is likely vaccination effects were not constant, as vaccines were initially prioritised for older individuals less likely to contribute to transmission.

## Supporting information

**S1 Table. Summary of REACT-1 testing numbers.** Number of swab-tests and positive swab-tests for each round of REACT-1
(XLSX)

**S2 Table. Summary of time-lags estimated from the segmented-exponential model.** Estimated time-lag parameter from segmented-exponential model fits to all data and subsets of the data.
(XLSX)

**S3 Table. Summary of discrete changes in the reproduction number estimated from the segmented-exponential model.** Estimated R for each time period and estimated multiplicative change in R for each change in restrictions from segmented-exponential model fits to all data and subsets of data.
(XLSX)

**S1 Fig. Smoothed estimates of infection prevalence by region.** Regional estimates of infection prevalence from 6 January to 12 July 2021 estimated using a Bayesian P-spline model fit to all 13 rounds of REACT-1 (only shown for rounds 8–13) assuming a constant second-order random-walk prior (value set from national model fit). All estimates of infection prevalence are shown with a central estimate (solid line) and 50% (dark shaded region) and 95% (light shaded region) credible intervals. Daily weighted estimates of swab positivity (points) are shown with 95% confidence intervals (error bars). Dashed lines show the date of key restriction changes in England.
(DOCX)

**S2 Fig. Smoothed estimates of infection prevalence by age group.** Prevalence estimates by age group from 6 January 2021 to 12 July 2021 estimated using a Bayesian P-spline model fit to all 13 rounds of REACT-1 (only shown for rounds 8–13) assuming a constant second-order random-walk prior (value set from national model fit). All estimates of infection prevalence are shown with a central estimate (solid line) and 50% (dark shaded region) and 95% (light shaded region) credible intervals. Daily weighted estimates of swab positivity (points) are shown with 95% confidence intervals (error bars). Dashed lines show the date of key restriction changes in England.
(DOCX)

**S3 Fig. Trends in the time-varying reproduction number assuming the Delta variant's generation time.** Rolling two-week average (averaged over prior two weeks) reproduction number as inferred from the Bayesian P-spline model fit to all data assuming a gamma distributed generation time with shape parameter = 2.20, and rate parameter = 0.48. Estimates of the reproduction number are shown with a central estimate (solid line) and 50% (dark shaded region) and 95% (light shaded region) credible intervals. The red line shows the probability that R>1 over time. Vertical dashed lines show the dates of key changes in restrictions. Horizontal dashed line shows R = 1 the threshold for epidemic growth.
(DOCX)

**S4 Fig. Regional estimates of the time-varying reproduction number.** Rolling two-week average (averaged over prior two weeks) reproduction number as inferred from the Bayesian P-spline models fit to data for each region assuming a gamma distributed generation time with shape parameter = 2.29, and rate parameter = 0.36. Estimates of Reproduction number are shown with a central estimate (solid line) and 50% (dark shaded region) and 95% (light shaded region) credible intervals. The red line shows the probability that R>1 over time. Vertical dashed lines show the dates of key changes in restrictions. Horizontal dashed line shows R = 1 the threshold for epidemic growth.
(DOCX)

**S5 Fig. Regional estimates of the time-varying reproduction number assuming the Delta variant's generation time.** Rolling two-week average (averaged over prior two weeks) reproduction number as inferred from the Bayesian P-spline models fit to data for each region assuming a gamma distributed generation time with shape parameter = 2.20, and rate parameter = 0.48. Estimates of Reproduction number are shown with a central estimate (solid line) and 50% (dark shaded region) and 95% (light shaded region) credible intervals. The red line shows the probability that R>1 over time. Vertical dashed lines show the dates of key changes in restrictions. Horizontal dashed line shows R = 1 the threshold for epidemic growth.
(DOCX)

**S6 Fig. Estimated discrete changes in the reproduction number by region and age group. (A, B)** Median estimates for R (points) and 95% credible intervals (error bars) for each period of time between changes in restrictions for the segmented-exponential model fit to subsets of data by region (A) and age-group (B) with the estimates obtained for all data also shown for comparison. **(C, D)** Median estimates for the multiplicative growth in R (points) and 95% credible intervals (error bars) for each step change in restrictions for the segmented-exponential model fit to subsets of data by region (C) and age-group (D) with the estimates obtained for all data also shown for comparison. In (A,C) Yorkshire is short for Yorkshire and The Humber.
(DOCX)

## Acknowledgments

We thank key collaborators on this work–Ipsos MORI: Kelly Beaver, Sam Clemens, Gary Welch, Nicholas Gilby, Kelly Ward and Kevin Pickering; Institute of Global Health Innovation at Imperial College: Gianluca Fontana, Didi Thompson and Lenny Naar; Molecular Diagnostic Unit, Imperial College London: Prof. Graham Taylor; Patient Experience Research Centre at Imperial College and the REACT Public Advisory Panel; NHS Digital for access to the NHS register; and the Department of Health and Social Care for logistic support.

## Author Contributions

**Conceptualization:** Oliver Eales, Christina Atchison, Graham Cooke, Wendy Barclay, Helen Ward, Ara Darzi, Deborah Ashby, Christl A. Donnelly, Paul Elliott, Steven Riley.

**Data curation:** Oliver Eales, Haowei Wang, David Haw, Kylie E. C. Ainslie, Caroline E. Walters.

**Formal analysis:** Oliver Eales.

**Funding acquisition:** Christina Atchison, Graham Cooke, Wendy Barclay, Helen Ward, Ara Darzi, Deborah Ashby, Christl A. Donnelly, Paul Elliott, Steven Riley.

**Methodology:** Oliver Eales.

**Software:** Oliver Eales.

**Supervision:** Steven Riley.

**Validation:** Oliver Eales.

**Visualization:** Oliver Eales.

**Writing – original draft:** Oliver Eales.

**Writing – review & editing:** Haowei Wang, David Haw, Kylie E. C. Ainslie, Caroline E. Walters, Christina Atchison, Graham Cooke, Wendy Barclay, Helen Ward, Ara Darzi, Deborah Ashby, Christl A. Donnelly, Paul Elliott, Steven Riley.

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
