## [Decision Letter · Decision Letter 0]

18 Aug 2022

Dear Mr Eales,

Thank you very much for submitting your manuscript "Trends in SARS-CoV-2 infection prevalence during England’s roadmap out of lockdown, January to July 2021" for consideration at PLOS Computational Biology.

As with all papers reviewed by the journal, your manuscript was reviewed by members of the editorial board and by several independent reviewers. In light of the reviews (below this email), we would like to invite the resubmission of a significantly-revised version that takes into account the reviewers' comments.

We cannot make any decision about publication until we have seen the revised manuscript and your response to the reviewers' comments. Your revised manuscript is also likely to be sent to reviewers for further evaluation.

Sincerely,

Benjamin Althouse

Academic Editor

PLOS Computational Biology

Tom Britton

Section Editor

PLOS Computational Biology

Reviewer's Responses to Questions

**Comments to the Authors:**

Reviewer #1: Authors analyse data from the REACT study in 2021 to show how transmissibility and infection prevalence increased during the re-opening phase. These data are extremely valuable and informative.

Major comments

1. You used a constant generation time distribution throughout, but actually there is good evidence that the generation time for Delta is slightly shorter. How would this impact your results?

Minor comments

Abstract line 43 “masked” may not be the best word, it was an intended benefit of vaccination at that time, perhaps a word like “offset” could be considered? On the other hand, Delta had increased intrinsic transmissibility. Maybe “also affected by …”

Line 101 — “ensured the results…” I’m not sure you can ensure the results are representative with a low response rate, but this is the best you could do to aim for representativeness.

Lines 169-170. Did you have information on vaccination status in your participants? This information would have been sufficient to use vaccination information in the regional model? With some general super-parameters for overall VE perhaps.

On terminology, “school closure” means pretty much everyone stays at home and the school gates are locked. For example a “snow day”. In UK wasn’t it more like “class dismissals” where staff went to the school and arranged online lessons, with most children staying at home but key worker children allowed into school campuses?

In Figure 3, how to interpret the two-week (prior two week) Rt, does it mean that the Rt on 29 March (transition to step 1b) reflects transmission in the last two weeks of March, or is it intended to be an estimate of the Rt on 29 March ? The latter would be easier to interpret, even if smoothed, because of the way you present the Rt against the policy changes. Specifically, does the bump in Rt on 29 March actually correspond to step 1b or is it a consequence of step 1a two weeks earlier?

Reviewer #2: The authors present a statistical analysis of REACT-1 COVID survey data in the UK, finding an increase in Rt following a relaxation of restrictions. The paper is well written, with clearly explained methods and informative figures. However, I think there are some ways in which the methodology could be strengthened.

1. I was surprised at the narrowness of the uncertainty intervals. This is perhaps most striking in Fig. 5, which has virtually zero uncertainty intervals over some periods, but also e.g. Fig. 3, where the notoriously hard-to-estimate reproduction number also has quite narrow uncertainty bands (such that many of the various wiggles seem to be statistically significant rather than just noise). I have full confidence that the authors have applied the statistical methods correctly, but I am concerned that they have not taken into account other sources of uncertainty.

2. For example, the authors note low response rate (11.7%), which could be a significant source of not just uncertainty, but bias (I note also that the swabs were self-administered, which could lead to underreporting). I wonder what Fig. 3 would look like with a different choice of window length (instead of 2 weeks). I wonder what the full ensemble of exponential fits would look like with varying delay lengths, rather than just the single delay shown in Fig. 5. (I also wonder how good the fits were with the segmented-exponential model; these are not shown, but my concern would be that the fewer segments used, the poorer the fit and yet paradoxically the smaller the uncertainty interval.) These methodological limitations would be unlikely to be resolvable within the REACT-1 dataset itself. Thus, the authors are encouraged to cross-validate these results by considering other data sources, such as case counts (which I agree on their own will have more bias than the REACT-1 data), hospital admissions (which will depend on variant), and other data sources (eg wastewater surveillance or seroprevalence). Finally, mobility data could be used to get an independent (and, yes, flawed) estimate of the expected impact of both the lockdowns and the relaxation.

3. In addition, I am not sure why the authors restricted the study to the dates they did. If REACT-1 was available from May 2020 to March 2022, why not use the whole time series? In particular, I would think the change in Rt from before to after the lockdown would be the single most valuable data point. As it is, this change happens before the analysis period, so its impact cannot be seen.

4. As it is, the results do not look, to my eye, especially convincing. Squinting at Fig. 3, Rt looks more or less flat from January through mid-May, then increases (but then starts to go down? -- using a longer time series would help). Consider the changes of restrictions as a scalar variable with 0 representing no interactions and 1 representing full interactions. Let's assume values could be estimated for this variable at different points in time (eg, by mobility data). Now consider the value of Rt at, say, 8 post-change as was done with the exponential model. If one then does a regression of these five values (with the restrictions/interactions variable on the x-axis and Rt on the y-axis), it's not clear to my eye that the correlation would be statistically significant. Put even more simply, I would expect the values in Fig. 5A to be monotonically increasing, but they are not.

In summary, while I think this is an interesting study, and while I think the content that is there is relatively strong, I feel a few additional data sources will be required in order to support the authors' claims that "the

lockdown was highly effective at reducing risk of infection". It is, if anything, surprising and a little disappointing that a clearer signal cannot be seen in the raw data.

**Have the authors made all data and (if applicable) computational code underlying the findings in their manuscript fully available?**

Reviewer #1: Yes

Reviewer #2: **No: **They point to the repository https://github.com/mrc-ide/reactidd, but this does not seem to cover the current paper.

PLOS authors have the option to publish the peer review history of their article (what does this mean?). If published, this will include your full peer review and any attached files.

Reviewer #1: No

Reviewer #2: No
---

## [Decision Letter · Decision Letter 1]

7 Nov 2022

Dear Mr Eales,

We are pleased to inform you that your manuscript 'Trends in SARS-CoV-2 infection prevalence during England’s roadmap out of lockdown, January to July 2021' has been provisionally accepted for publication in PLOS Computational Biology.

Best regards,

Benjamin Althouse

Academic Editor

PLOS Computational Biology

Tom Britton

Section Editor

PLOS Computational Biology

Reviewer's Responses to Questions

**Comments to the Authors:**

Reviewer #1: No further comments

**Have the authors made all data and (if applicable) computational code underlying the findings in their manuscript fully available?**

Reviewer #1: Yes

PLOS authors have the option to publish the peer review history of their article (what does this mean?). If published, this will include your full peer review and any attached files.

Reviewer #1: No

---

## [Editor Report · Acceptance letter]

18 Nov 2022

PCOMPBIOL-D-22-00962R1 

Trends in SARS-CoV-2 infection prevalence during England’s roadmap out of lockdown, January to July 2021

Dear Dr Eales,

I am pleased to inform you that your manuscript has been formally accepted for publication in PLOS Computational Biology. Your manuscript is now with our production department and you will be notified of the publication date in due course.

With kind regards,

Zsofi Zombor
